# Microfluidic droplet application for bacterial surveillance in fresh-cut produce wash waters

J. Brian Harmon[1,2]*, Hannah K. Gray[1], Charles C. Young[1,2], Kellogg J. Schwab[1]

1 Department of Environmental Health and Engineering, Johns Hopkins Bloomberg School of Public Health, Baltimore, Maryland, United States of America, 2 Asymmetric Operations Sector, Johns Hopkins University Applied Physics Laboratory, Laurel, Maryland, United States of America

* jbrianharmon@gmail.com

**Data Availability Statement:** Relevant data are within the paper and supplemental files. Complete data sets are publicly available at: https://doi.org/10.5061/dryad.djh9w0vx1.

## Abstract

Foodborne contamination and associated illness in the United States is responsible for an estimated 48 million cases per year. Increased food demand, global commerce of perishable foods, and the growing threat of antibiotic resistance are driving factors elevating concern for food safety. Foodborne illness is often associated with fresh-cut, ready-to-eat produce commodities due to the perishable nature of the product and relatively minimal processing from farm to the consumer. The research presented here optimizes and evaluates the utility of microfluidic droplets, also termed ultra-miniaturized bioreactors, for rapid detection of viable *Salmonella enterica* ser. Typhimurium in a shredded lettuce wash water acquired from a major Mid-Atlantic produce processing facility (denoted as Producer) in the U.S. Using a fluorescently-labeled anti-*S*. Typhimurium antibody and relative fluorescence intensities, paired with in-droplet incubation, *S*. Typhimurium was detected and identified with 100% specificity in less than 5 h. In initial optimization experiments using *S*. Typhimurium-spiked sterile water, the relative fluorescence intensity of *S*. Typhimurium was approximately two times that of the observed relative intensities of five non-*S*. Typhimurium negative controls at 4-h incubation in droplets containing Rappaport-Vasiliadis (RV) broth at 37˚C: relative fluorescence intensity for *S*. Typhimurium = 2.36 (95% CI: 2.15–2.58), *Enterobacter aerogens* 1.12 (95% CI: 1.09–1.16), *Escherichia coli 700609* = 1.13 (95% CI: 1.09–1.17), *E. coli 13706* 1.13 (95% CI: 1.07–1.19), *E. coli 700891* 1.05 (95% CI: 1.03–1.07) and *Citrobacter freundii* 1.04 (95% CI: 1.03–1.05). *S*. Typhimurium- and *E. aerogens*-spiked shredded lettuce wash waters acquired from the Producer were then incubated 4 h in-droplet at 37˚C with RV broth. The observed relative fluorescence of *S*. Typhimurium was significantly higher than that of *E. aerogens*, 1.56 (95% CI: 1.42–1.71) and 1.10 (95% CI: 1.08–1.12), respectively. While further optimization focusing on compatible concentration methodologies for highly-dilute produce water samples is needed, this application of droplet microfluidics shows great promise in dramatically shortening the time necessary–from days to hours–to confirm viable bacterial contamination in ready-to-eat produce wash waters used throughout the domestic and international food industry.

**Funding:** This work was supported by grants from the National Institute for Occupational Safety and Health (T42 OH0008428) (https://www.cdc.gov/niosh/index.htm) to the Johns Hopkins University Education and Research Center for Occupational Safety and Health, the National Institutes of Health (T32 ES007141) (https://www.nih.gov/). The funders had no role in study design, data collection and analysis, decision to publish, or preparation of the manuscript. The Osprey Foundation of Maryland. The funders had no role in study design, data collection and analysis, decision to publish, or preparation of the manuscript. We would like to thank the major Mid-Atlantic ready-to-eat produce processing facility (denoted as Producer) for providing wash water. The funders had no role in study design, data collection and analysis, decision to publish, or preparation of the manuscript.

**Competing interests:** The authors have declared that no competing interests exist.

## Introduction

Foodborne illness is a significant public health concern worldwide, with a global burden of disease comparable to HIV/AIDS, malaria and tuberculosis [1]. The World Health Organization estimated there were approximately two billion cases and over one million deaths associated with foodborne illness in 2010 [2]. According to the Centers for Disease Control and Prevention (CDC), approximately 48 million cases of foodborne illness occur annually in the United States (U.S.). [3]. Trend analyses for reported foodborne illness in the U.S. from 1996 to 2013 suggests infections from bacterial pathogens commonly associated with foodborne illness are remaining relatively constant (i.e. there is a lack of evidence showing reduction) [4]. While Norovirus is the leading cause of foodborne illness in the U.S. with over five million cases per year, non-typhoidal *Salmonella enterica* is the second leading cause of foodborne illness in the U.S. with over one million cases per year and is estimated to cause more hospitalizations and deaths than any other foodborne pathogen [5]. The ready-to-eat food industry is particularly susceptible to pathogen contamination along the farm-to-consumer route due to minimal processing, plant tissue damage encouraging microbial growth, and the perishable nature of fresh produce necessitating rapid delivery to the consumer [6]. Elevated incidence of foodborne illness linked to produce has been reported in the U.S., Canada, and European Union [7,8]. During the 1998 to 2008 reporting period, nearly half (46%) of all foodborne illnesses in U.S. were attributable to produce, with 22% attributable to leafy greens alone [9]. Given the global threat of foodborne contamination, the food industry must continually evaluate critical control points (CCPs) for its most vulnerable crops, improve upon antiquated detection methods, and maintain a collaborative relationship with national and international surveillance networks [10–12].

The food industry utilizes the hazards analysis and critical control point (HACCP) framework as a tool to mitigate foodborne hazards, analyze ways and means to counter hazards, identify CCPs, and routinely evaluate the effectiveness of the control strategies to mitigate hazards [13]. The ready-to-eat, fresh-cut produce industry has numerous CCPs along the farm-to-consumer route, including irrigation and wash water, handling, and shelf-life [14–16]. A HACCP-based plan, utilizing foodborne outbreak surveillance data to ensure microbial food safety should be an integral part of local, regional, and national produce processing facilities [12,17].

Fresh-cut and leafy green produce wash waters play an intricate role in the industrial processing of ready-to-eat vegetables, including removing debris from plant surfaces and providing microbial disinfection via chlorination [18]. Produce wash waters are also a well-documented CCP in the ready-to-eat produce industry, and can potentially cross-contaminate otherwise safe produce when disinfection efficacy is absent or hazards are not mitigated [19–22]. Potential for cross-contamination is a notable hazard at this CCP as chlorine residual can decay or become inactivated by organic material loading in the wash water [22,23]. Given the challenges posed and risks associated with contamination from wash waters, researchers are actively exploring methods to increase food safety associated with fresh-cut produce wash waters. Using a semi-commercial pilot scale system, researchers at the U.S. Department of Agriculture explored the use of an additive, T128, to chlorinated produce wash water. This additive was designed to increase antimicrobial efficacy and further reduce cross-contamination, as well as reduce bacterial pathogen survival [24].

Microfluidic droplet technology extends the scientific capacity for exploring chemical and biological systems by improving throughput and resolution, at lower cost and higher sensitivity than conventional methods. Droplet single-cell confinement enables rapid cell identification and phenotypic and genotypic characterization of diverse cell populations. Microfluidics

allow for generation and evaluation of millions of cells within hours, as compared to conventional culture-based approaches limited by growth rates, plate size, appropriate dilution and cell density, and inability to culture certain environmental bacteria [25,26]. Droplet microfluidic tools include metabolite identification, probing for unique cell types, and other cell-based and novel applications [27,28]. Cells in droplets can be quantified and characterized using fluorescent activated cell sorting, droplet optical density measurement and Raman spectroscopy [29–31].

These techniques are applied to rapid antibiotic resistance and pathogen detection, probing rare and difficult-to-culture bacteria, accessing unique metabolisms and cell processes like biodegradation and substrate utilization, all within heterogenous bacterial communities [32–34]. Conventional analytical tools can be extended to droplet microfluidics, like droplet-based mass spectrometric analysis, which is being used to identify cell production of antibiotic metabolites within droplets at a single-droplet level [35]. Advancements in single-cell drop sequencing support whole-genome sequencing, RNA sequencing, and gene screening on single cells to bridge the gap between rare phenotypes and genotype [28,36,37].

Applying in-droplet microfluidic technologies, and creating ultra-miniaturized bioreactors, can bridge the gap between prolonged viable culture methods and rapid non-viable culture-independent methods. Micron-scale microfluidic droplets have a volume of only pico- to nano-liters, and can form an ultra-miniaturized bioreactor for a single bacterial cell. Researchers have already utilized in-droplet microfluidics to isolate *E. coli* in a pico-liter water-in-oil bioreactor and used time-lapse fluorescent imaging to study metabolic activity using 4-methy-lumbelliferyl-β-D-glucuronide (MUG), a florigenic reporter [38]. Marcoux and colleagues reported fluorescent confirmation of encapsulated *E. coli* growth and metabolism in less than 2 h [38]. Recently, chemotactic microbes, such as *E. coli*, were sorted from both mixed and pure cultures using a chemoeffector concentration gradient and manipulating the bacterial cell motility away from the gradient [39]. Innovative microfluidic technologies have also been used to assess and identify antibiotic resistance among bacterial populations [40]. An in-droplet microfluidic platform was developed to screen bacteria for antibiotic resistance in clinically-relevant isolates, and responses from the bacteria and rapid characterization was achieved within one hour [41].

This study showcases the potential for microfluidic droplets to serve both as a means of rapid detection and viable culturing capability, and reports the first known evaluation of in-droplet microfluidics as a tool for rapidly ($\leq$ 5 h) and specifically identifying *S*. Typhimurium in produce (i.e. shredded lettuce) wash water acquired from a real-world Producer. Initially, specificity of the fluorescein isothiocyanate- (FITC)-labeled antibody targeting the common surface antigen of *S*. Typhimurium was evaluated. In-droplet growth experiments with *S*. Typhimurium, and five negative controls, were then conducted in Rappaport-Vassiliadis (RV) selective broth medium with the *S*. Typhimurium-specific FITC antibody (FITC-Ab). Once specificity and growth were characterized across the bacterial panel, the real-world shredded lettuce wash water was filtered to remove plant debris, spiked with $10^6$ CFU/ml *S*. Typhimurium, encapsulated in droplets, and analyzed by fluorescent detection in RV broth medium selective for *S*. Typhimurium (Fig 1).

## Results

All bacterial species under study (a positive control and five negative controls) were consistently replicated with expected lag and doubling times in tryptic soy broth (TSB) (Table 1). TSB was utilized as a positive control for the medium variable under test, and was expected to promote growth of *S*. Typhimurium and non-*S*. Typhimurium. TSB growth curves also served

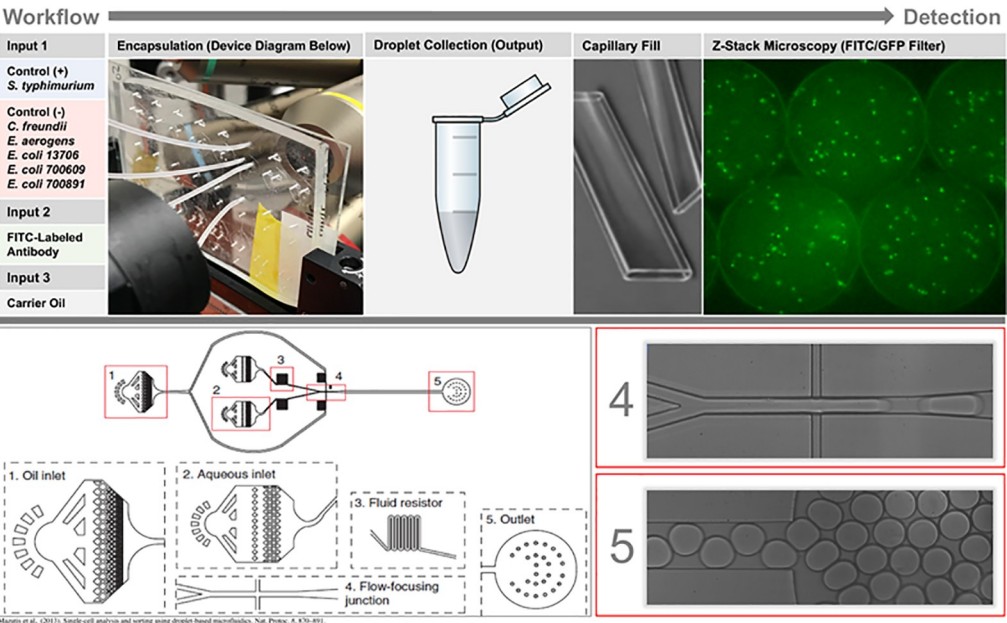

**Fig 1. Workflow diagram highlighting inputs, output, sample preparation, and fluorescence procedure.** A previously published diagram is utilized paired with real-time, high-speed photography of droplet formation (4) and collection at output (5).

as the standard for generational growth, or doubling time, across all organisms examined. Generation time across all species in TSB ranged from 23.1 to 39.6 minutes (Table 1). Tryptic soy media are a known universal growth medium for bacteria in microbiological laboratories [42]. Its use in this study should not be interpreted as useful for suppressing growth of the negative control bacterial species. All test bacteria exhibited some measure of growth in buffered peptone water (BPW), with generation times ranging from 29.3 to 203.9 minutes (Table 1). The range of generation times across the bacterial species is an indicator that BPW either suppresses growth of many non-*S.* Typhimurium, or does not provide the ideal nutrition for some bacterial species under test (Table 1).

The RV broth medium at 37˚C promoted desired growth of *S.* Typhimurium and inhibited the growth of four out of five non-*S.* Typhimurium negative controls. However, one non-*S.*

**Table 1. Generation time[a] and lag phase duration[b] of the tested bacteria in different culture broth medium (TSB, BPW and RV broth) at 37˚C or 41.5˚C (RV broth only), with an initial bacterial concentration of $10^3$ CFU/ml medium.**

| Bacterial Strain | TSB 37˚C | BPW 37˚C | RV 37˚C | RV 41.5˚C |
|---|---|---|---|---|
| *S.* Typhimurium | 37.5 (4) | 51.4 (5.5) | 39.8 (6) | 91.2 (<1) |
| *C. freundii* | 39.6 (6) | 66.7 (6.5) | 103.5 (9.5) | No growth |
| *E. aerogens** | 23.1 (3.75) | 35.2 (3.5) | 49.2 (7.5) | No growth |
| *E. coli 700609* | 28.7 (5.5) | 60.8 (5.5) | No growth | No growth |
| *E. coli 13706* | 28.8 (5) | 42.5 (6) | No growth | No growth |
| *E. coli 700891* | 24.8 (4) | 38.7 (4.25) | No growth | No growth |

a: Log phase doubling time listed in minutes;

b: Lag phase duration listed in h and in parentheses;

* Optimal growth temperature for E. aerogens is 30˚C; TSB = Tryptic Soy Broth, BPW = Buffered Peptone Water, RV = Rappaport-Vassiliadis Broth, PBS = Phosphate Buffered Saline

Typhimurium negative control, *E. aerogens*, was able to replicate in the RV medium. The generation times for *S.* Typhimurium and *E. aerogens* were 39.8 and 49.2 minutes, respectively (Table 1). RV broth proved highly effective at eliminating or drastically suppressing growth of non-*S.* Typhimurium species and encouraging the growth of only *S.* Typhimurium at 37˚C. Of note, the ideal growth temperature for this formulation of RV, like most RV formulations, is 41.5˚C ± 5. RV broth medium at 41.5˚C, the ideal growth temperature for this formulation of RV, significantly suppressed the growth rate of the *S.* Typhimurium strain under test, however, this temperature completely inhibited the growth of all five negative controls (Table 1). All negative controls were inhibited at 41.5˚C, however, the generation time for *S.* Typhimurium at 41.5˚C was deemed too low to successfully reach a detectable threshold of bacterial cells in ≤ 5 h. Furthermore, this research is intended to be translatable to in-droplet microfluidic growth and detection. It was anticipated that droplet integrity could be compromised at prolonged exposure to 41.5˚C.

Fluorescent contribution of the evaluated media under test were also measured along with varying concentrations of the FITC-Ab. TSB exhibited the highest level of auto-fluorescence. RV broth exhibited the least auto-fluorescence, and the intensity of FITC-Ab fluorescence was reduced by approximately a $\log_{10}$ RFU (Fig 2; Table 2).

Bacteria at a $10^8$ CFU/ml concentration were encapsulated in-droplet in combination with a FITC-Ab concentration of 1.66 µg/ml. Using 1,000:250:250 µl/h flow rates and 50/50 merging parameters, each droplet contained approximately 100 bacterial cells and 0.83 µg/ml FITC-Ab. This approach ensured each droplet had ample bacteria to assess the degree of specificity for *S.* Typhimurium and cross-reactivity with negative controls. Both sample inputs utilized PBS as the dilution solution in order to best assess FITC-Ab specificity for *S.* Typhimurium and potential cross-reactivity, or lack of specificity, with non-*S.* Typhimurium species. *S.* Typhimurium had the highest RFU measured, however, *E. coli* 13706 exhibited a high propensity for cross-reactivity. The two other *E. coli* strains, 700609 and 700891, showed moderate cross-reactivity. *C. freundii* had minimal cross-reactivity with the FITC-Ab (Fig 3).

Microscopic fluorescent images, with embedded metadata for fluorescent intensity, were performed through the entirety of the droplet at five micron intervals. Once merged into a two-dimensional image, these data were used as the raw data for relative fluorescence measurements (Figs 3 and 4).

Once a baseline for *S.* Typhimurium specificity and potential cross-reactivity of the FITC-Ab was achieved in PBS (Figs 3 and 4), RV broth was evaluated in-droplet for growth suppression of non-*S.* Typhimurium bacteria. Bacteria at a $10^6$ CFU/ml concentration in autoclaved deionized water were encapsulated in-droplet along with a FITC-Ab concentration of 20 µg/ml. Using 1,000:250:250 µl/h flow rates and 50/50 merging parameters, one in every two-to-ten droplets contained a single, isolated bacterial cell and 10 µg/ml FITC-Ab. Using this approach ensured growth in-droplet originated from a single bacterial cell to assess the observed growth kinetics and changes in relative fluorescence over time. Data for each time point, stratified by bacterial species/strain, are reported in Supplemental Information (SI). Relative fluorescence data for the five-hour incubation are also reported in SI.

Microscopic fluorescent images, with embedded metadata for fluorescent intensity, were performed through the entirety of the droplet at five micron intervals. These data were collected at each time point of incubation for each bacterial species/strain under test. Once merged into a two-dimensional image, these data were used as the raw data for relative fluorescence measurements in Ni-E Elements software (Figs 5 and 6).

*S.* Typhimurium and *E. aerogens* were subsequently spiked into shredded lettuce wash water collected at the Producer's facility. The wash water physiochemical properties at time of collection were 17.05 mg/L organic chloramine, 50.5 nephelometric turbidity units, 4.37 pH

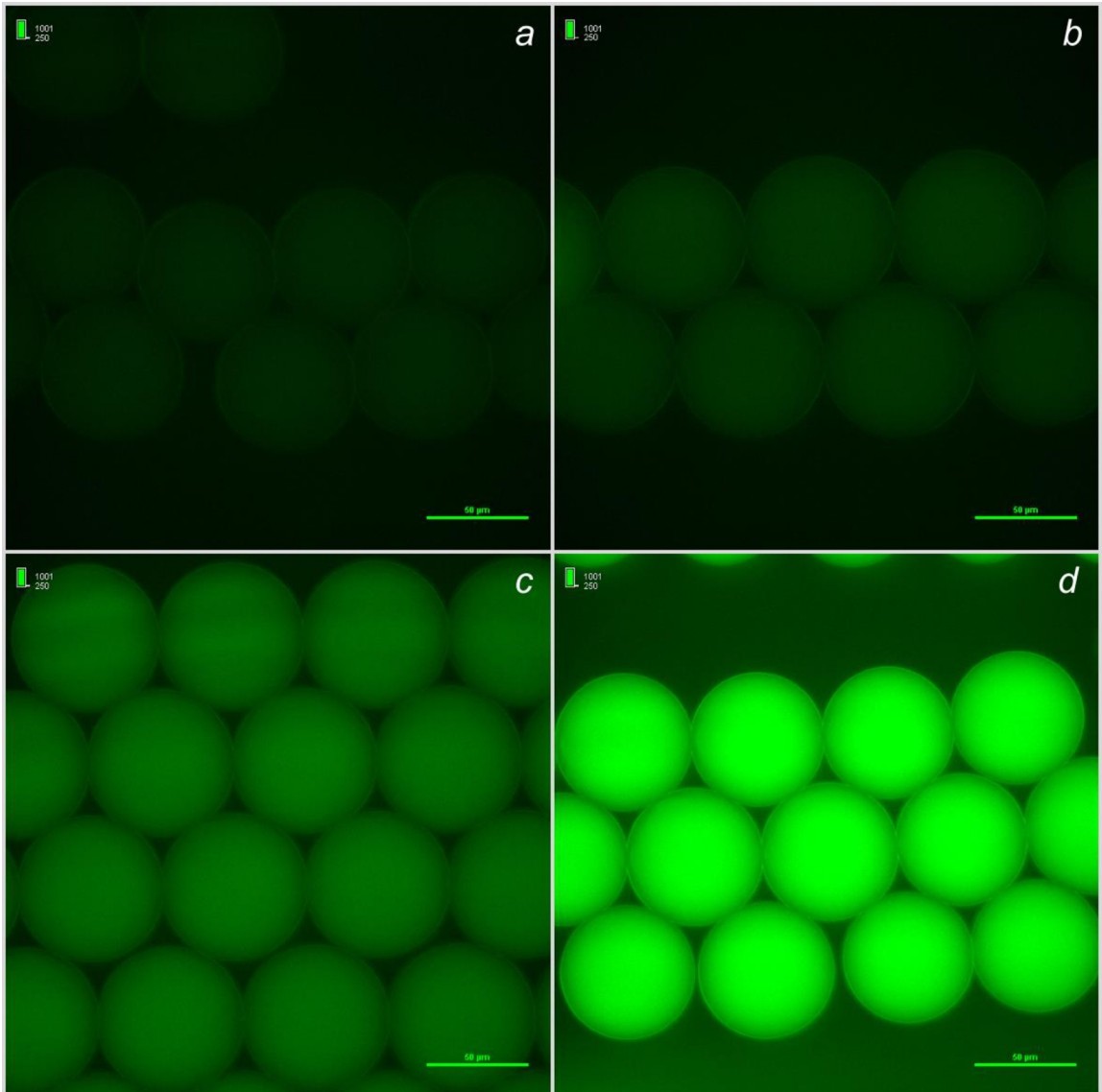

**Fig 2. Medium and FITC-Ab were encapsulated in 50–70 μm diameter droplets (1x medium and 0.83 μg/ml FITC-Ab in-droplet) to visually assess medium contribution to fluorescence.** Fluorescence exposure and look-up-table (LUT) settings are standardized across all four media evaluated. The LUT values can be found in the upper left-hand corner of each image. Media under test included phosphate buffered saline (a), Rappaport-Vassiliadis (b), buffered peptone water (c), and tryptic soy broth (d).

(after thawing from stock, 5.97 pH), and a conductivity of 1,008 (mS/cm). Any remaining chlorine (organic chloramines) were quenched by 1 mM sodium thiosulfate, and the wash water was filtered to remove any debris 0.22 μm or larger. The target bacterial spiked concentration was $10^6$ CFU/ml for both *S.* Typhimurium and *E. aerogens*. This bacterial cell concentration ensures there is approximately one bacterial cell encapsulated every two to ten droplets formed. Relative fluorescence of the encapsulated bacteria and growth kinetics at 37˚C from 0 to 5 h incubation were measured (Fig 7). Fluorescent images with embedded metadata were used to acquire relative fluorescence via Ni-E Elements software and visual confirmation of growth detection for *S.* Typhimurium (Figs 7, 8 and 9).

**Table 2. Fluorescence and pH of medium commonly used for traditional culture of _S._ Typhimurium.**

| Mediumᵃ | Control [pH]ᵇ | 0.1 µg/ml FITC-Abᶜ | 1 µg/ml FITC-Abᶜ | 10 µg/ml FITC-Abᶜ |
|---|---|---|---|---|
| _TSB_ | 151.8 [7.57] | 167.9 (16.1) | 296.6 (144.8) | 1,527.4 (1,375.6) |
| _BPW_ | 31.7 [7.30] | 43.1 (11.3) | 157.2 (125.5) | 1,306.5 (1,274.7) |
| _RV_ | 26.2 [5.13] | 29.3 (3.1) | 44 (17.8) | 207.9 (181.7) |
| _PBS_ | 1.0 [7.75] | 10.1 (9.1) | 137.3 (136.3) | 1,565.6 (1,564.6) |

a: TSB = Tryptic Soy Broth, BPW = Buffered Peptone Water, RV = Rappaport-Vassiliadis Broth, PBS = Phosphate Buffered Saline

b: Control fluorescence intensities used for normalization, measured pH value of medium in brackets, 0 µg/ml FITC-Ab in Control;

c: Mean relative fluorescence units of four replicates represented; Normalized values in parentheses

Published research has reported faster bacterial growth rates for bacteria incubated inside droplets, when compared to growth rates using traditional methods [43]. Separate tubes (one tube per each time point) were also collected in parallel for both _S._ Typhimurium and _E. aerogens_ incubation experiments in order to perform aerobic spread plate counts and analyze observed growth rate in-droplet. Droplets were ruptured using PFO releasing the trapped incubated content, diluted accordingly, spread on TSA plates, and incubated over night at 37°C [29]. Given PFO was introduced upon completion of incubation in the droplet, effects of PFO on bacterial growth were not examined. Aerobic plate counts indicated both _S._ Typhimurium and _E. aerogens_ were in lag phase from 0 to approximately 2 h in-droplet. Both _S._ Typhimurium and _E. aerogens_ initiated exponential growth at 2 h and replicated exponentially

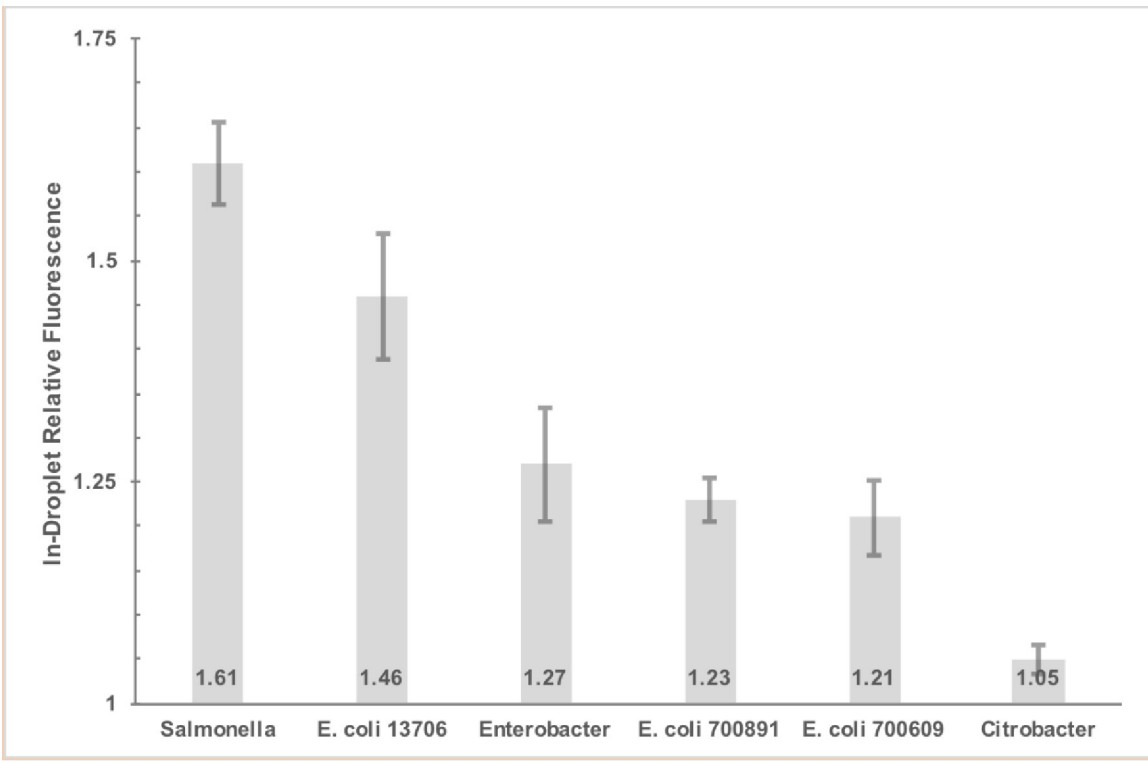

**Fig 3. Fluorescence of bacteria relative to background fluorescence in-droplet of six bacterial species encapsulated in droplet with 0.83 µg/ml FITC-Ab per droplet in phosphate buffered saline.** Five fluorescence measurement replicates were performed per bacterial species/strain. Error bars represent the 95% confidence interval.

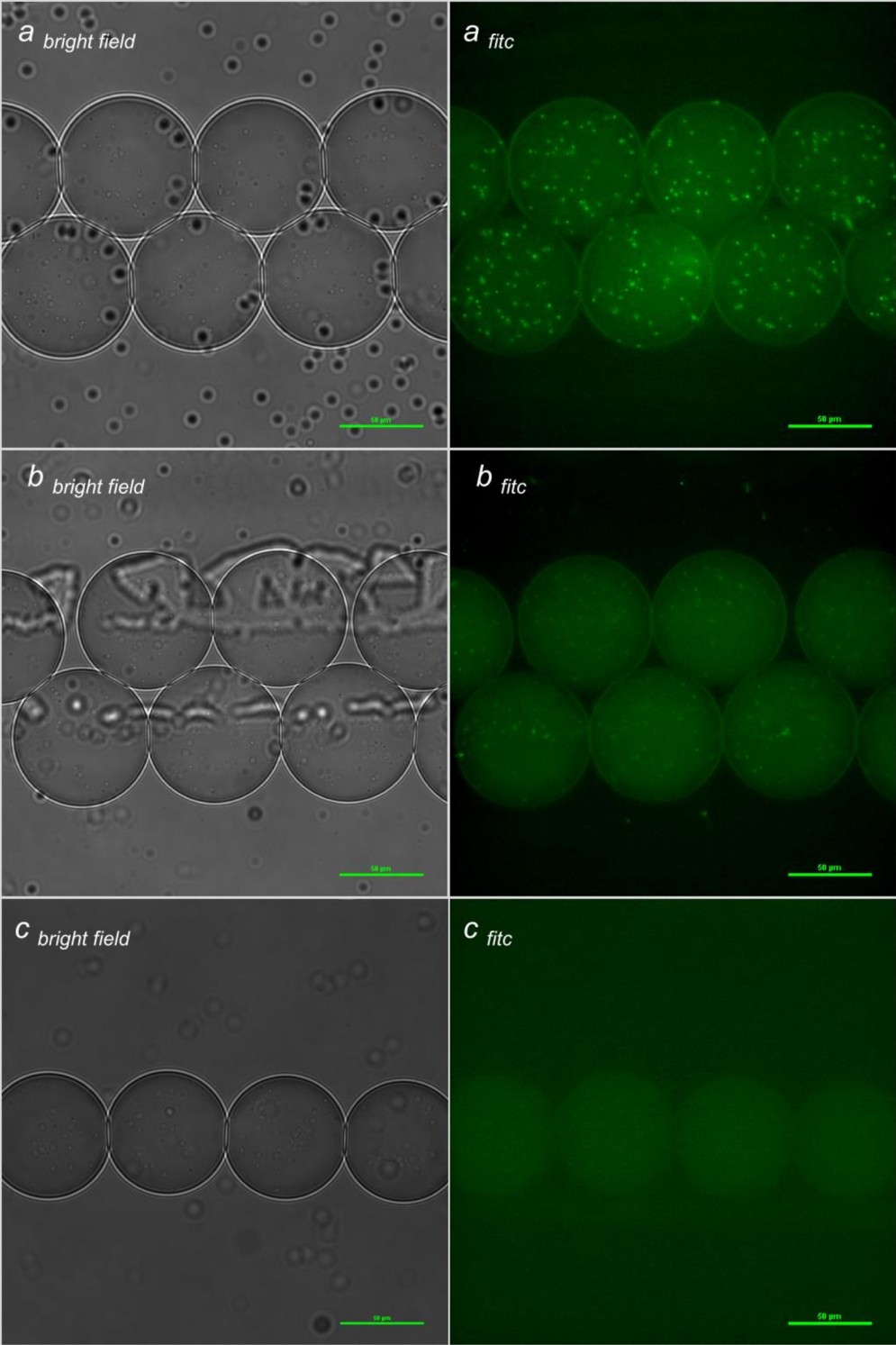

**Fig 4. Bacterial species (*a*: *S.* Typhimurium*; b*: *E. aerogens; c*: *C. freundii*) with 0.83 µg/ml FITC-Ab in phosphate buffered saline.) each droplet is approximately 50–70 µm in diameter (scale on image is 50 µm).** Bright field images are presented complimentary to FITC images and visually show bacterial concentration at an ideal focal plane (5 µm interval stacking and merging of images into two dimensional images is not optimal for bright field images). FITC images are merged 5 µm interval focal plane images and represent a two-dimensional image of the entire droplet.

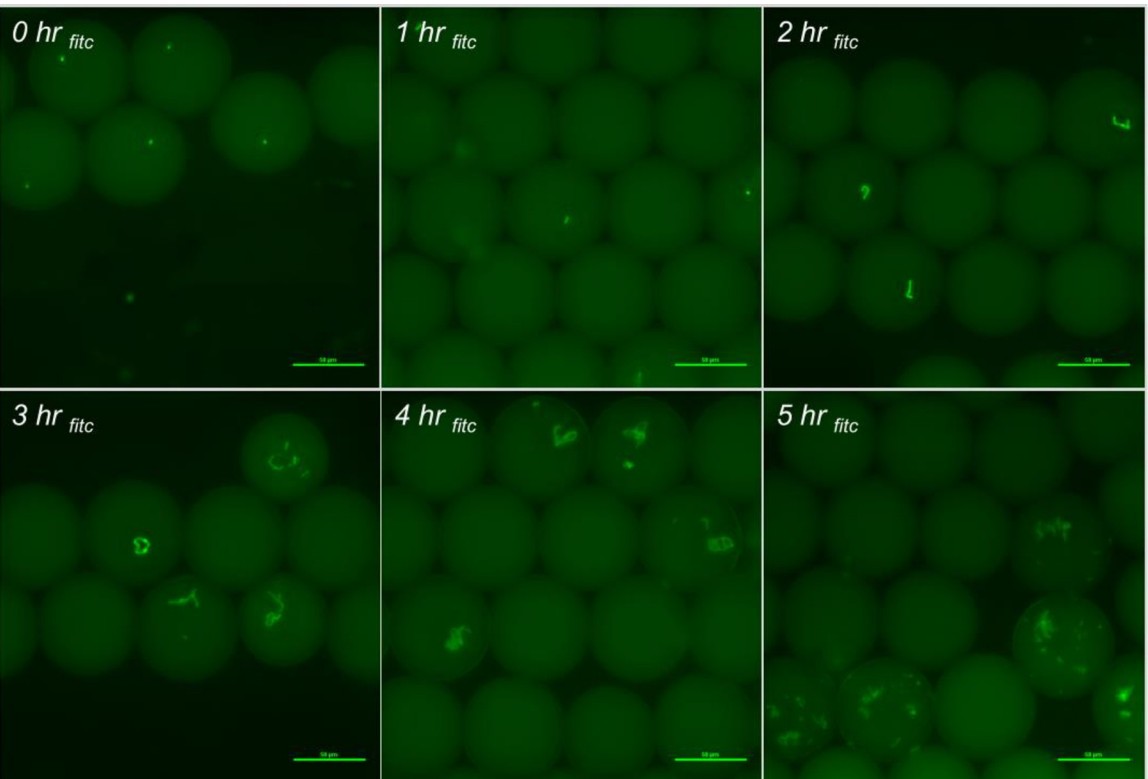

**Fig 5. Fluorescent images of *S*. Typhimurium incubation in-droplet with sterile deionized water, and 10 μg/ml FITC-Ab in 1x RV broth at 37˚C.**

until hour four of incubation. Observed lag times and $\log_{10}$ growth phase generation time in-droplet using aerobic plate counts are summarized and compared to observed lag times and $\log_{10}$ growth (Table 3).

## Discussion

### Ultra-miniaturized bioreactors show promise for bacterial surveillance

The RV broth, at both temperatures under test (37 and 41.5˚C), effectively eliminated or drastically suppressed growth of non-*S*. Typhimurium species (Table 1). In traditional microbiological culturing using a selective medium, RV broth at 41.5˚C would be superior in selecting for *S*. Typhimurium among the panel tested. Even at the sub-optimal temperature of 37˚C, *S*. Typhimurium likely would be able to out-compete the only other bacteria species identified in this study that grow at a suppressed rate in RV broth, *E. aerogens* and *C. freundii*. In contrast to traditional selective methods, the objective of this study was to assess the feasibility for RV broth use in a microfluidic droplet. Only the *S*. Typhimurium under test successfully grew in RV broth at 41.5˚C, preventing growth of all non-*S*. Typhimurium under test. At first glance, 41.5˚C would be the temperature of choice given complete suppression of all non-*S*. Typhimurium species. However, the growth rate of *S*. Typhimurium was substantially reduced at 41.5˚C, when compared to *S*. Typhimurium grown in RV at 37˚C (Table 1). Using *S*. Typhimurium with a generation time of 91.2 minutes at 41.5˚C as an example (Table 1), and assuming each droplet is starting with one *S*. Typhimurium cell, it would take 9 h to reach approximately 60 *S*. Typhimurium cells per droplet and 12 h to reach approximately 250 *S*.

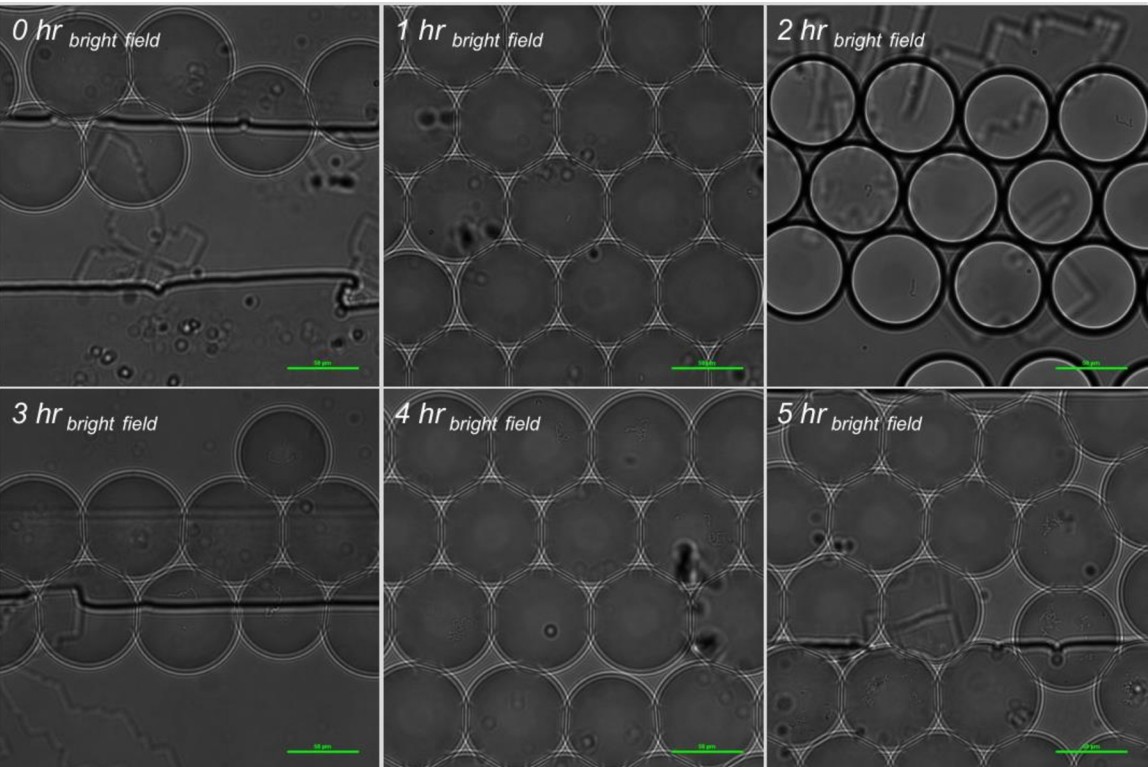

**Fig 6. Bright field images of *S*. Typhimurium incubation in-droplet with sterile deionized water, and 10 μg/ml FITC-Ab in 1x RV broth at 37˚C.**

Typhimurium cells per droplet. For overnight culture purposes, the reduced generation time of RV at 41.5˚C is not an issue of concern, since 18-to-24 h is utilized with inoculation of more than one *S*. Typhimurium cell. For this microfluidic droplet application, RV broth at an incubation temperature of 37˚C was selected as the objective is to detect *S*. Typhimurium growth in 4–8 h.

As demonstrated by the collected data, a more prolific bacterial growth was supported during incubation in RV broth at 37˚C as compared to 41.5˚C. However, RV at 37˚C does not suppress growth of both *C. freundii* and *E. aerogens*. In order to maximize selective growth rates in RV at 37˚C, a fluorescently-labeled antibody specific for *S*. Typhimurium to increase specificity was utilized in conjunction with RV broth to screen out *C. freundii* and *E. aerogens*. Given fluorescently-labeled antibodies can create a relatively high detectable signal in the presence of their target antigen, and the multitude of fluorophores available for researchers to conjugate to their antibody of choice, this method proved to be the ideal way moving forward in creating a double positive indicator for *S*. Typhimurium. Masking of fluorescence must be taken into account, as these data indicate fluorescent intensities exhibit a $log_{10}$ reduction in RFU when diluted in RV broth (Table 2; analysis reported in SI). The reduced fluorescence intensity observed in RV broth is likely due to a pH of 5.3. Published findings have shown FITC to be pH-sensitive, exhibiting reduced fluorescence signal at a low pH [44,45]. It may be possible to increase specificity with an alternate fluorophore-antibody conjugate.

There are other fluorescent options commercially available for *S*. Typhimurium detection, however, not all options are applicable inside a droplet. Two of the more applicable fluorescent methods would be either a fluorescent medium or fluorescently-labeled antibody for *S*.

**Fig 7. Relative fluorescence of encapsulated *S.* Typhimurium and *E. aerogens* in shredded lettuce wash water (0.5x in-droplet) and incubated in Rappaport-Vassiliadis broth (1x in-droplet) for 5 h with a FITC-Ab concentration of 10 µg/ml in-droplet.** A potential detection threshold region is identified on the figure. No measurable relative fluorescence was identified in time points 0 through 2 for both *E. aerogens*. Five replicates were measured for each bacterial species at each time point.

Typhimurium. There is one fluorescent substrate, 4-methylumbelliferyl caprylate (MUCAP), for *S.* Typhimurium [46]. However, MUCAP is non-soluble in water, and thus is not compatible for aqueous in-droplet applications. For the purposes of this research, a fluorescently-labeled antibody specific for *S.* Typhimurium was a sound course of action when used in tandem with RV broth in-droplet. A direct binding antibody, such as an antibody targeting *S.* Typhimurium's common surface antigen, could prove effective for detection purposes. A fluorescent indicator for detection purposes would be an ideal indicator for *S.* Typhimurium detection in-droplet, and a FITC-Ab could be applied in future microfluidic applications. It would be necessary to evaluate medium contributions to fluorescence in-droplet to best assess potential uses in the future. This research evaluated the impact medium can have on overall fluorescence (Fig 2; Table 2).

An additional aspect to consider when developing a method to rapidly detect *S.* Typhimurium would be incubation in-droplet. Some incubation would be necessary in order to reach

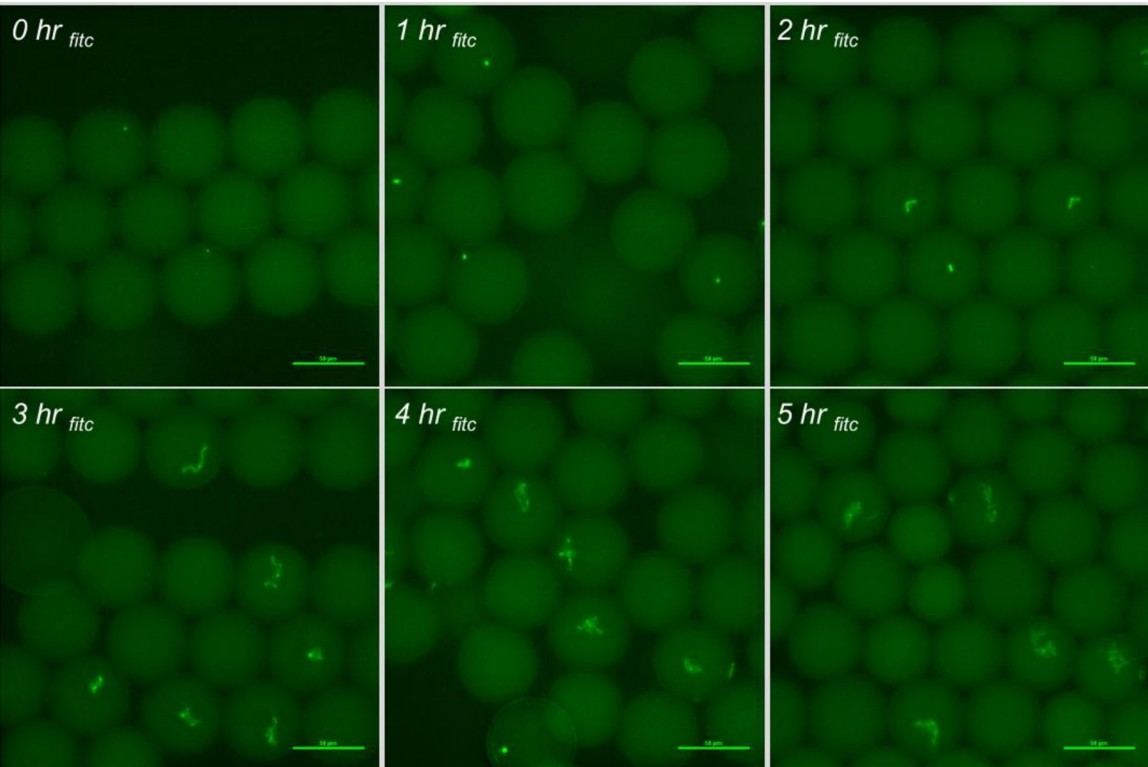

**Fig 8. Fluorescent images of *S.* Typhimurium incubation in-droplet with 0.5x shredded lettuce wash water, and 10 μg/ml FITC-Ab in 1x RV broth at 37˚C.**

optically detectable levels of colorimetric or fluorescence indicators. Current enumerating technologies are incapable of reliably detecting a single *S.* Typhimurium cell encapsulated inside a droplet on an enumeration or sorting microfluidic chip. This may seem to be a limiting factor, and in some ways, it is not ideal. However, using the selective properties of the RV broth can further help specificity of differentiating *S.* Typhimurium from non-*S.* Typhimurium in mixed samples. Current capabilities of the in-droplet microfluidic system necessitate an incubation time to achieve a detectable in-droplet bacterial cell concentration for enumeration or potentially sorting of the target bacterial organism in the future. The incubation step also differentiates viable versus non-viable cells as there is potential non-viable bacterial cells would still be capable of binding an antibody. The incubation step ensures single bacterial cells do not yield a false positive. By determining how many h it takes to achieve a detectable number of *S.* Typhimurium, in comparison to the non-*S.* Typhimurium, increased specificity can be achieved. For example, according to these data, *C. freundii* doubling time in RV broth at 37˚C is over two times that of the *S.* Typhimurium strain under examination in this study (Table 1). The specificity of isolation and detection of *S.* Typhimurium can potentially be achieved given the suppressed growth state of *C. freundii* and relatively high replication rate of S. Typhimurium at 37˚C. The *E. aerogens* performed considerably well in the RV broth at 37˚C, and only had doubling times approximately 10 minutes longer than the *S.* Typhimurium strain under test in this study (Table 1). Therefore, given the narrow temperature window between 37˚C and 41.5˚C, it would be worthwhile for future research to address varying concentrations of mixed cultures with *S.* Typhimurium and non-*S.* Typhimurium, such as *C. freundii* and *E. aerogens*.

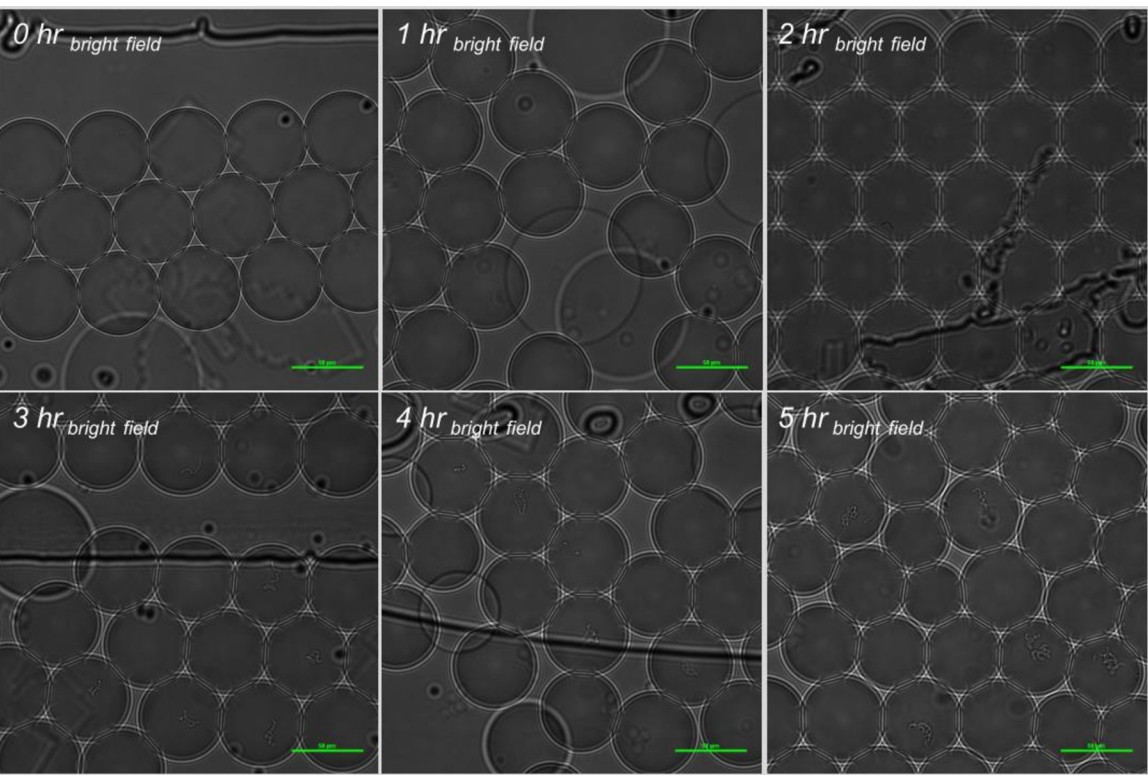

**Fig 9. Bright field images of _S._ Typhimurium incubation in-droplet with 0.5x shredded lettuce wash water, and 10 μg/ml FITC-Ab in 1x RV broth at 37˚C.**

Finally, the U.S. Food and Drug Administration Bacteriological Analytical Manual (FDA BAM) states RV broth is selective for _S._ Typhimurium, and the data provided in this study reaffirms FDA guidance. The results also show RV broth has a precise temperature window and can be less selective if used at the incorrect temperature, i.e. less than 41.5˚C. When used as directed at 41.5˚C for incubation, RV broth is effective at suppressing non-_S._ Typhimurium growth. However, generation times are increased greatly for _S._ Typhimurium. It has already been noted _S._ Typhimurium would likely outcompete another serotype of bacteria in a mixed culture–at least the bacterial serotypes researched in the study. A more accurate determination for the concentration of _S._ Typhimurium can be achieved by carrying out future experiments of mixed cultures, and more effective selective media for _S._ Typhimurium could potentially be

**Table 3. Comparison of lag phase duration and generation times for _S._ Typhimurium and _E. aerogens_ outside and inside droplet.**

| Bacteria | Generation and Lag Time, Non-Droplet [a] | Generation and Lag Time, In-Droplet [b] |
|---|---|---|
| _S._ Typhimurium | 40[c] (6)[d] | 27 (1–2) |
| _E. aerogens_ | 49 (7.5) | 21 (1–2) |

a: Turbidity/absorbance used for observed lag time and generation time, 200 μl of $10^3$ CFU/ml starting concentration;

b: Aerobic spread plate count used for observed lag time and generation time, 1 bacterial cell per 50–70 μm droplet;

c: Generation time in minutes;

d: Observed lag time in h

formally tested, certified, and eventually used by the scientific community. It is important to note that tetrathionate (TT) broth, which has been reported as an additional optional selective broth for *S.* Typhimurium by the FDA BAM [47], was not examined in this study due to the presence of a calcium carbonate precipitate that does not fully dissolve when the medium is prepared resulting in subsequent clogging or fouling of the microfluidic channels on-chip.

Initial in-droplet encapsulation experiments focused on assessing specificity of the FITC-Ab for *S.* Typhimurium, and the cross-reactivity with non-*S.* Typhimurium species. Potential medium contribution to fluorescence were eliminated by utilizing PBS for these experiments. Moreover, by suspending both bacterial cells and FITC-Ab in PBS, a more precise ratio of bacterial fluorescence to background could be assessed. A relatively high concentration of bacterial cells (approximately 100 cells) was generated in-droplet to better visualize cross-reactivity of the FITC-Ab with negative controls, and to provide ample sites for relative fluorescence measurements. The anti-*S.* Typhimurium FITC-Ab exhibited good sensitivity by binding to all encapsulated *S.* Typhimurium via *S.* Typhimurium's CSA-1, and displayed robust relative fluorescence values (Fig 3). Deficiencies in specificity, however, are exposed by evaluating calculated relative fluorescence values and fluorescence images for the negative control panel (Fig 3). Cross-reactivity with the FITC-Ab and members of the negative control panel was anticipated as *E. coli*, *E. aerogens*, and *S.* Typhimurium are similar in many ways. All three are gram-negative, from the *E. aerogensiaceae* family, and can be found in the gastrointestinal tract of an animal host [48]. For this reason, use of a *S.* Typhimurium selective broth, RV, was then utilized to suppress growth of non-*S.* Typhimurium species in-droplet.

Assessment and evaluation of media identified by the FDA BAM was carried out, and RV broth was identified as the best candidate for in-droplet use to detect *S.* Typhimurium. Two key aspects must be noted for the use of RV broth in this study. RV broth exhibits fluorescence masking of the FITC-Ab, and this masking reduces relative fluorescence units by approximately one $\log_{10}$ (reported in SI). Therefore, FITC-Ab concentrations were increased from 0.83 μg/ml to 10 μg/ml for in-droplet RV broth experiments carried out in this study. Also, *S.* Typhimurium had a much higher generation rate in RV broth at 37˚C instead of the manufacturer-directed incubation temperature of 41.5˚C (RV R10 Broth, Becton Dickinson) (Table 1).

RV broth performed as expected in suppressing most non-*S.* Typhimurium species under test, based on experiments outside of droplets. While growth was not completely inhibited, all *E. coli* species were noticeably suppressed by the presence of RV broth (reported in SI). One *E. coli* strain evaluated, ATCC 700891, was particularly suppressed. For instance, relative measurements were unattainable at time points 0 through 2 h. The FITC-Ab cross-reacted minimally with *C. freundii* in both PBS and RV broth with sterile deionized water. This is a promising observation as *C. freundii* is one of two bacterial species evaluated capable of growth in RV broth at 37˚C.

In experiments utilizing PBS and RV broth in sterile deionized water, the FITC-Ab had moderate-to-high cross reactivity with *E. aerogens* (reported in S5 Fig). Moreover, *E. aerogens* grows well in RV broth at 37˚C (Table 1). For this reason, *E. aerogens* was selected to be tested as the negative control when spiking the real-world shredded lettuce wash water acquired from the Producer. Fluorescent images with embedded metadata were captured every h and analyzed for a 5-h incubation of both *S.* Typhimurium and *E. aerogens* (Figs 7 and 8; *E. aerogens* images reported in S14 and S15 Figs). The wash water conditions under test were intended to simulate a highly contaminated sample, and encapsulate one bacterial cell per two-to-ten droplets generated. This allowed for monitoring of growth kinetics in-droplet over the five-hour time course. Both *S.* Typhimurium and *E. aerogens* had reduced relative fluorescence values in the real-world wash water experiments (Fig 7) when compared to experiments utilizing RV broth and sterile deionized water (S5 Fig).

The manufacturer of the RV broth formulation utilized in this study states the pH is 5.1 at 25°C (Becton Dickinson). Moreover, it has been reported that FITC fluorescence intensity has a direct correlation with pH gradients. Ma et al. explored the relationship between decreasing FITC intensity with decreases in pH [45]. Utilizing the figures reported by Ma et al., there is nearly a $log_{10}$ reduction in fluorescence intensity of FITC between a 7.5 pH solution and a 5.0 pH solution. This fluorescence relationship with pH could help explain the masking effect observed from RV broth when compared to other media (Table 2). A pH-related FITC intensity relationship is also something to consider in relation to the shredded lettuce wash water. The pH of the shredded lettuce wash water was 4.37 upon collection, and measured to have a pH of 5.97 after thawing from stock. A 20 ml volume of 50/50 2x RV broth and shredded lettuce wash water mixture (simulating in-droplet concentrations) was measured and had a pH of 5.26. This low pH likely explains the reduction in fluorescence intensity observed in the wash water experiments.

Bacterial growth in-droplet was also evaluated and compared to growth outside of droplets. Both *S*. Typhimurium and *E. aerogens* exhibited faster generation times and reduced observed lag times in-droplet (Table 3). More rapid droplet-based incubation growth rates have been reported in earlier studies, and is thought to be due to rapid mixing, reduced contaminants, reproducibility of droplets, and small volumes [43]. Lag times were minimal in-droplet as well. *S*. Typhimurium and *E. aerogens* initiated log phase growth between 1 and 2 h of incubation. These observed increased growth rates, and decreased lag times, further showcase microfluidic droplets as a rapid application for viable bacterial detection. It should be noted growth rates outside of droplets were calculated using turbidity measurements for growth, and these samples under test did not contain FITC-Ab.

Of note, this rapid microfluidic detection capability will need to be paired with a bacterial concentration technique. Recent published research highlights a range of concentration techniques that are potentially compatible with the droplet microfluidic system, such as continuous-flow centrifugation and ultrafiltration [49–52]. Follow-on laboratory testing and evaluation to determine the most suitable sample concentration technique is necessary to maximize the potential for this surveillance technique.

## Why advancing food surveillance matters

Foodborne illness is an ever-present international threat to public health [1]. In the U.S. alone, 1 in 6 people will acquire foodborne illness annually and this national burden is estimated to cost between $55.5 and $93.2 billion [53]. Fresh-cut produce and leafy greens are often implicated in foodborne outbreaks, and it has been estimated leafy greens are the most common cause of foodborne outbreaks in the U.S. [54]. Developing and improving upon methods for surveillance and detection of food contamination that can be implemented in food industry HACCP control strategies are necessary to ensure safer food supplies in the future [13,55]. Moreover, developing novel methodologies to isolate, detect, and prevent bacterial outbreaks are essential to reduce mortality, illness, hospitalization, associated cost to taxpayers and overall international burden [56]. When deployed systematically with emerging engineering technologies, applications using in-droplet microfluidics show great promise for rapid, same-day/same-shift, and viable bacterial detection at CCP's, such as fresh-cut produce wash waters.

## Materials and methods

### Bacterial strains used, preparation, and protocols

A total of six ATCC bacterial strains, one *S*. Typhimurium positive control and five negative controls, were utilized (Table 4). All strains used were Biosafety Level 1 due to the microfluidic

**Table 4. Bacterial strains utilized in this study.**

| Bacterial Strain | ATCC ID | Other Designation(s) | Biosafety Level |
|---|---|---|---|
| *C. freundii* (Braek) Werkman and Gillen | 8090 | NCTC 9750 | 1 |
| *E. aerogens* Homaeche and Edwards | 13048 | NCDC 819–56, NCTC 100X | 1 |
| *Escherichia coli* (Migula) Castellani and Chalmers | 13706 | CIP 104337, NCIB 12416 | 1 |
| *Escherichia coli* (Migula) Castellani and Chalmers | 700609 | CN13 | 1 |
| *Escherichia coli* (Migula) Castellani and Chalmers | 700891 | Hs(pFamp)R | 1 |
| *S. typhimurium enterica* subsp. *enterica* (ex Kauffmann and Edwards) Le Minor and Popoff serovar Typhimurium | 53647 | Chi4062 | 1 |

platform location outside of a biological safety containment hood. Stocks were generated from overnight culture and stored at -20˚C. Working stocks at $10^9$ colony forming units (CFU)/ml in phosphate buffered saline (PBS) were stored at 5˚C. Streak plates for isolated colonies were performed for each experiment on tryptic soy agar plates (bioMerieux) and incubated overnight at 37˚C. Isolated colonies from overnight incubation were then diluted into PBS until an absorbance of 0.05 ± 0.005 at 660 nm, equating to $10^9$ CFU/ml. Following an absorbance confirmation of 0.05 ± 0.005 at 660 nm, a 1:10 dilution series was carried out to reach $10^8$ and $10^6$ CFU/ml, the concentrations ideally suited for microfluidic flow rates of encapsulation for control panel and Rappaport-Vassiliadis (RV) growth experiments, respectively. Control panel experiments utilized a bacterial input of $10^8$ CFU/ml. This bacterial input concentration, paired with microfluidic flow rates used, yielded approximately 100 bacterial cells encapsulated per droplet. RV broth growth kinetic experiments utilized a bacterial sample input of $10^6$ CFU/ml. This bacterial input concentration, paired with microfluidic flow rates used, yielded approximately one bacterial cell encapsulated every two-to-ten droplets. Bacterial dilutions and spread plate counts on tryptic soy agar plates (bioMerioux) were performed in shredded lettuce wash water growth experiments with *S.* Typhimurium and *E. aerogens* to identify length of lag phase, initiation of log growth phase, and generations time in minutes. Droplets with encapsulated bacteria were incubated at 37˚C for the desired time and then droplets were lysed using 1H, 1H, 2H, 2H-Perfluoro-1-Octanol (PFO) (Sigma). Approximately 10–20 μl of PFO was used per sample with brief centrifugation. The amount of PFO used varied slightly based on volume of droplets captured. Once popped, the bacterial cells in PFO form an immiscible layer on top of the medium. An initial 1:100 dilution was performed due to the small volume of cells in PFO, and subsequent 1:10 dilutions were performed to achieve the desired spread plating CFU/ml concentration. Samples from time points zero and one h were assayed by spread plate in duplicate, and time points two through five were assayed by spread plate in triplicate.

## Growth conditions, equipment, and measurement

Bacteria were grown for 24 h at either 37˚C or 41.5˚C (RV only) and turbidity measurements were taken every 15 minutes using an automated absorbance device (BioScreen C, Automated Microbiology Growth Curve Analysis System, Growth Curves USA, New Jersey, United States). Given one objective of this research was to examine the limiting capability of RV medium for non-*S.* Typhimurium bacterial species, all samples were grown at 37˚C–the optimum growth temperature for the *S.* Typhimurium utilized. The control panel was also grown at 41.5˚C in the RV broth for 24 h, per medium manufacturer instructions. Of note, optimal growth temperature for the *E. aerogens* strain used is 30˚C. The starting cell concentration for

all bacterial species was $10^3$ colony-forming unit CFU/ml and all bacterial species/strains studied were performed in triplicate.

Each bacterial species/strain used in this study was cultivated from stocks by an overnight culture. Streak plates for isolation were then performed for each bacterial species and allowed to grow overnight. Isolated bacterial colonies were then added to PBS until an absorbance of $0.05 \pm 0.005$ at 660nm was reached. An additional absorbance, $0.1 \pm 0.05$ at 600nm, was measured to ensure equipment was working properly. Starting with a $10^9$ CFU/ml suspension in PBS a 1:10 dilution series was repeated until reaching a $10^5$ CFU/ml concentration in PBS. Spread plating for CFU/ml counts were performed on tryptic soy agar plates (bioMerieux) in duplicate during this experiment to ensure CFU/ml concentrations were accurate, and all concentrations were found to have a starting concentration of $10^9$ CFU/ml. A 1:10 dilution series was performed in order to achieve $10^5$ CFU/mL concentration in PBS. A 1:100 dilution was then performed by transferring 10 µl of $10^5$ CFU/ml into 990 microliters of medium. A 1:100 dilution was performed at this stage to minimize the input of PBS, and thus not diluting the medium under test. A 200 microliter aliquot of $10^3$ CFU/mL bacterial concentration in medium were then added per well of the proprietary Bioscreen Honeycomb 100-well plate. Three replicates were performed per bacterial species/strain in respective medium.

## Determination of generation time

Growth curves were generated for each bacterial strain, and stratified by each of the four conditions under test. The slope of the bacterial log-phase growth was utilized for the calculation of generation time. Length and slope of the log-phase varied across all bacteria. To calculate the slope, a log-linear trend line was fit to the log-phase, and an $r^2$ and natural log equation for each condition under test was attained.

## Medium and antibody preparation, and pH measurements

RV broth was prepared at 2x concentration by adding double the manufacturer-directed amount in grams per unit volume (Beckton-Dickinson). Once mixed in filtered deionized water, the 2x RV broth mixture was autoclaved at 121°C for 15 minutes. An affinity purified polyclonal fluorescein isothiocyanate- (FITC)-conjugated antibody (Ab) targeting the common surface antigen- (CSA)-1 of *S*. Typhimurium was utilized for activated fluorescence detection (BacTrace, Anti-*S*. Typhimurium, CSA-1 Antibody, FITC-Labeled, SeraCare). FITC-Ab from stocks (0.5 mg/ml in PBS, 10 µl aliquots) were added to 2x RV broth to prepare a 20 µg/ml FITC-Ab concentration in 2x RV broth. A 2x RV broth with 20 µg/ml FITC-Ab was used for input on-chip as the concentration per droplet would be reduced by half upon droplet formation and bacterial encapsulation using the co-flow on-chip device, i.e. 1x RV broth with 10 µg/ml FITC-Ab upon droplet formation and bacterial encapsulation. The pH of RV broth medium was measured using a pH meter (VWR Symphony SB70P Digital, Bench-model pH Meter).

## Fluorescent antibody and fluorescent measurement procedures

A FITC-conjugated antibody (FITC-Ab), anti-*S*. Typhimurium, CSA-1 Antibody (BacTrace), was utilized to measure medium (TSB, BPW, and RV) contribution to overall fluorescence at concentrations of 0.1, 1, and 10 µg/ml. Measurements were taken by fluorimeter (Tecan) in a 96-well black bottom plate at 488 nm excitation and 525 nm emission. The FITC-Ab was also merged into droplets at a 1:300 concentration from FITC-Ab stocks (0.5 mg/ml). Once merged into the droplet, at a 50/50 ratio with the complimentary input sample, the ratio in-droplet was 1:600 FITC-Ab (0.83 µg/ml in-droplet FITC-Ab concentration).

## Wash water collection, characterization, and preparation

Wash water from a shredded lettuce processing line was acquired from a major ready-to-eat vegetable processing facility on the East Coast. Initial samples were divided into 500 ml aliquots and stored at -80˚C. Working stocks were prepared in aliquots of 5 ml and stored at -20˚C. Collected wash waters for all vegetable samples were analyzed for free chlorine and organic chloramines by the N,N-diethyl-*p*-phenylene diamine colorimetric method [57]. Turbidity, conductivity, and pH were also measured by Hach 2100N Turbidimeter (Hach Company, CO), Hach Sension5 portable conductivity meter (Hach Company, CO) and accutupH$^+$ probe (Fisher Scientific), respectively. Once thawed from stocks, pH of the shredded lettuce wash water was measured using a VWR pH meter (VWR Symphony SB70P Digital, Bench-model pH Meter). For microfluidic experiments, the shredded lettuce wash water was filtered through a 0.22 μm syringe filter (Durapore PVDF Membrane, MILLEX GV) and spiked with bacteria to an approximate concentration of 10$^6$ CFU/ml. Residual chlorine was quenched with 1 mM sodium thiosulfate (Sigma).

## Microfluidic consumables, equipment, and procedure

The co-flow microfluidic device utilized in this research was designed at Harvard University, and published in Nature Protocols [29]. Microfluidic chips used in this research were fabricated by the Research and Exploratory Development Department (REDD), Johns Hopkins University Applied Physics Laboratory (JHUAPL). Microfluidic equipment, consumables, and technical support were supplied by the Applied Biological Sciences Group, JHUAPL. The microfluidic droplet system utilizes a proprietary surfactant (Phasex Corp.) at a concentration of 646 mg per 40 ml 7500 Novec Oil (3M). This surfactant-oil mixture aided in oxygen transfer, served as the carrier medium and facilitated uniform droplet formation of the sample input(s) on-chip. Flow rates reported in the Nature Protocol were 180:90:90 μl/h (2:1:1 flow rate ratio) for the surfactant-oil mixture and the two sample inputs [29]. Flow rates utilized in this research were modified from 2:1:1 to 2:0.5:0.5 (surfactant-oil to sample(s) input flow rate). Slight modifications were made for uniformity of observed droplet diameter upon formation during experiments. The on-chip device used in this research had one surfactant-oil input and two sample inputs. If two sample inputs are utilized, the overall input flow rate of the samples must be split between the two inputs. For example, if flow rate of the surfactant-oil input is 2,000 μl/h, sample input A will be 500 μl/h and sample input B will be 500 μl/h. Flow rates for this research were typically 1,000 μl/h surfactant-oil input, and 250 μl/h for each sample input. Samples were loaded into a syringe using a blunt fill needle (BD Blunt Fill Needle). The fill needle was then removed and replaced with a precision needle (BD Precision Glide). Tubing was connected between the sample syringe and the microfluidic device on-chip (95 Durometer LDPE, Scientific Commodities Micro Medical Tubing, 0.015" I.D x 0.043" O.D.). Proprietary apparatus pumps designed for syringes (1 ml to 30 ml volume BD Luer Lock Tip) were used to regulate flow rates (Harvard Apparatus). Pump flow rate was managed via a software program compatible with Harvard Apparatus (LabVIEW 16, National Instruments). Droplet generation was monitored in real-time using Nikon objectives equipped with a high-speed camera and complimentary software (FASTEC IL5). Droplets were captured in 1.5 ml experimental tubes (Eppendorf).

## Microscopy equipment and procedures

Formed droplets were drawn into 50 μm diameter glass capillary tubing for visualization and measurement (0.05 x 0.5 mm ID, VitroTubes). Filled capillaries were then fixed to a microscope slide (Fischer) by first sealing the loading end and elevating the slide at a 45-degree angle

for 1–5 minutes. This allowed the droplets to spread out equally and stack for imaging. The non-loading end was then sealed before microscopy. The Eclipse Ni-E motorized microscope system and Ni-E Analysis Elements software (Nikon) equipped with a digital camera (ORCA-Flash 4.0 LT, Hamamatsu) were used for fluorescent images and data analysis. Permission to utilize the Eclipse Ni-E motorized microscope system and software was granted by REDD, JHUAPL. Z-stack data at five μm intervals was collected throughout entirety of the droplet using the ND Acquisition tool in Ni-E Analysis Elements. Droplet diameter ranged from approximately 50–70 μm. Z-stack data were then merged into flat, data-rich images using the EDF tool in General Analysis, Ni-E Analysis Elements. Merging Z-stack data allowed for maximum image capture of encapsulated bacteria by creating a two-dimension data image of bacteria at differing focal planes. Point estimates of fluorescence intensity were gathered and a complimentary fluorescence intensity of background was paired with each measurement. Fluorescence measurements were normalized by reporting relative fluorescence. Relative fluorescence was calculated by dividing bacterial fluorescence by background fluorescence (*Fluorescence$_{bacteria}$ / Fluorescence$_{background}$*). A sample size of five (n = 5) relative fluorescence measurements were used for all mean, standard deviation, and 95% confidence interval calculations. Fluorescent data images were smoothed and normalized using Ni-E Elements software in order to report comparable, high resolution images.

## Supporting information

**S1 Table. Relative fluorescence of bacteria to background in-droplet grown in Rappaport-Vassiliadis broth over 5-hour incubation at 37˚C.**
(DOCX)

**S1 Fig. TSB = Tryptic Soy Broth; BPW = Buffered Peptone Water; RV = Rappaport-Vassiliadis Broth; PBS = Phosphate Buffered Saline.** Direct fluorescence measurements were taken at 0.1, 1, and 10 μg/ml FITC-Ab in each media with four replicates. The grey trend line represents the normalized fit for each media to better display fluorescent contribution of media.
(DOCX)

**S2 Fig. TSB = Tryptic Soy Broth; BPW = Buffered Peptone Water; RV = Rappaport-Vassiliadis Broth; PBS = Phosphate Buffered Saline.** Adjusted fluorescence measurements of 0.1, 1, and 10 μg/ml FITC in each media with four replicates. The grey trend line represents the PBS control to better display fluorescent contribution of media, and the fluorescent masking effect of RV and/or pH-sensitivity of FITC at low pH.
(DOCX)

**S3 Fig. Direct fluorescence intensity measurements and relative fluorescence to phosphate buffered saline (PBS) of each media-FITC-Ab data image in Fig 6.** A total of five fluorescence values (n = 5) were analyzed for each media and the PBS control. Medias include Rappaport-Vassiliadis broth (RV), buffered peptone water (BPW), and tryptic soy broth (TSB). Direct fluorescence measurements correspond with the left y-axis, and relative fluorescence measurements correspond with the right y-axis. Error bars represent the 95% confidence interval.
(DOCX)

**S4 Fig.  Bacterial strains (a: *E. coli* 700609; b: *E. coli* 13706; c: *E. coli* 700891) with 0.83 μg/ml FITC-Ab in phosphate buffered saline.) each droplet is approximately 50–70 μm in diameter (scale on image is 50 μm).** Bright field images are presented complimentary to

FITC images and visually show bacterial concentration at an ideal focal plane (5 μm interval stacking and merging of images into two dimensional images is not optimal for bright field images). FITC images are merged 5 μm interval focal plane images and represent a two-dimensional image of the entire droplet.
(DOCX)

**S5 Fig. Relative fluorescence of bacterial species (*S*. Typhimurium, *E. aerogens*, and *C. freundii*) and strains (*E. coli* 700609, 13706, and 700891) incubated in Rappaport-Vassiliadis broth for five hours at 37˚C with a FITC-Ab concentration of 10 μg/ml in-droplet.** The trend line represents a decrease in relative fluorescence of *S*. Typhimurium over the time course. A detection threshold region is identified on the figure. No measurable relative fluorescence was identified in time points 0 through 2 for both *C. freundii* and *E. coli* 700891. Five replicates were measured for each bacterial species/strain.
(DOCX)

**S6 Fig. Fluorescent images of *E. aerogens* incubation in-droplet with sterile deionized water, and 10 μg/ml FITC-Ab in 1x RV broth at 37˚C.**
(DOCX)

**S7 Fig. Bright field images of *E. aerogens* incubation in-droplet with sterile deionized water, and 10 μg/ml FITC-Ab in 1x RV broth at 37˚C.**
(DOCX)

**S8 Fig. Fluorescent images of *E. coli* 700609 incubation in-droplet with sterile deionized water, and 10 μg/ml FITC-Ab in 1x RV broth at 37˚C.**
(DOCX)

**S9 Fig. Bright field images of *E. coli* 700609 incubation in-droplet with sterile deionized water, and 10 μg/ml FITC-Ab in 1x RV broth at 37˚C.**
(DOCX)

**S10 Fig. Fluorescent images of *E. coli* 13706 incubation in-droplet with sterile deionized water, and 10 μg/ml FITC-Ab in 1x RV broth at 37˚C.**
(DOCX)

**S11 Fig. Bright field images of *E. coli* 13706 incubation in-droplet with sterile deionized water, and 10 μg/ml FITC-Ab in 1x RV broth at 37˚C.**
(DOCX)

**S12 Fig. Images of *E. coli* 700891 incubation in-droplet with sterile deionized water, and 10 μg/ml FITC-Ab in 1x RV broth at 37˚C.** No relative fluorescence detection at hours 0 through 2.
(DOCX)

**S13 Fig. Images of *C. freundii* incubation in-droplet with sterile deionized water, and 10 μg/ml FITC-Ab in 1x RV broth at 37˚C.** No relative fluorescence detection at hours 0 through 2.
(DOCX)

**S14 Fig. Fluorescent images of *E. aerogens* incubation in-droplet with 0.5x shredded lettuce wash water, and 10 μg/ml FITC-Ab in 1x RV broth at 37˚C.**
(DOCX)

**S15 Fig. Bright field images of *E. aerogens* incubation in-droplet with 0.5x shredded lettuce wash water, and 10 μg/ml FITC-Ab in 1x RV broth at 37˚C.**
(DOCX)

## Author Contributions

**Conceptualization:** J. Brian Harmon, Hannah K. Gray, Charles C. Young, Kellogg J. Schwab.

**Data curation:** J. Brian Harmon, Charles C. Young, Kellogg J. Schwab.

**Formal analysis:** J. Brian Harmon, Charles C. Young, Kellogg J. Schwab.

**Funding acquisition:** Charles C. Young, Kellogg J. Schwab.

**Investigation:** J. Brian Harmon, Charles C. Young, Kellogg J. Schwab.

**Methodology:** J. Brian Harmon, Hannah K. Gray, Charles C. Young, Kellogg J. Schwab.

**Project administration:** Kellogg J. Schwab.

**Resources:** Charles C. Young, Kellogg J. Schwab.

**Software:** J. Brian Harmon.

**Supervision:** Charles C. Young, Kellogg J. Schwab.

**Validation:** J. Brian Harmon, Kellogg J. Schwab.

**Visualization:** J. Brian Harmon, Kellogg J. Schwab.

**Writing – original draft:** J. Brian Harmon, Hannah K. Gray, Charles C. Young, Kellogg J. Schwab.

**Writing – review & editing:** J. Brian Harmon, Hannah K. Gray, Charles C. Young, Kellogg J. Schwab.

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
