## [Decision Letter · Decision Letter 0]

19 Feb 2020

PONE-D-19-34504

Microfluidic droplet application for bacterial surveillance in fresh-cut produce wash waters

PLOS ONE

Dear Dr. Harmon,

Thank you for submitting your manuscript to PLOS ONE. After careful consideration, we feel that it has merit but does not fully meet PLOS ONE’s publication criteria as it currently stands. Therefore, we invite you to submit a revised version of the manuscript that addresses the points raised during the review process.

ACADEMIC EDITOR: 

The editor would like to apologize to the authors for the long duration of the review process; it was due to the unavailability of multiple reviewers with expertise in the field which were approached and invited by the editor to handle the manuscript but could not accept the invitation. Review comments have been timely provided by only one invited reviewer and are attached in the present letter. Given the considerable delay already encountered, and in an effort to provide the authors with a timely decision, this manuscript proceeded to the next stage of the editorial process, based on the evaluation provided by one reviewer and the academic editor.

Although the academic editor is not an expert in microfluidic applications, raised comments regarding the overall approach of the presented study from a food microbiology perspective, its scientific soundness as well as the writing quality of the manuscript are summarized below. According to the editor’s opinion, the manuscript should be thoroughly revised, mainly in terms of structure and presentation, before its publication in PLOS ONE is considered. In its current form, the manuscript is rather wordy and hard-to-follow at several points, making it hard for the reader to extract and focus on its most important and novel findings.

- The “Introduction” section is very long for a research paper; only in review articles is justified for such section to be wordy and actually include subsections. Please reduce significantly the length of the introduction and revise its content so that excessive details are avoided and a focus is placed on the information providing the rationale for conducting this particular study. Some of the information currently present in the introduction would better fit in the “Discussion” section of the manuscript, with the latter, however, also being rather wordy and requiring significant revision.

- L24: please revise to “ready-to-eat produce commodities”

- L27: revise to “*Salmonella enterica* ser. Typhimurium” (since it is the first time the bacterial species is mentioned and its full name should be presented); upon first mentioning, you can directly use “*S.* Typhimurium” for the rest of the manuscript. Please make sure that genus and species are in italics, whereas serotype is capitalized and not in italics.

- L31 and throughout the text of the manuscript: use “h” instead of “hours”

- L34: please revise to “4-h incubation”

- L40: correct to “…incubated for 4 h…”

- L35-38: are these numbers referring to the relative fluorescence intensity? Please rephrase for clarification.

- L53-55: please consider revising to “According to the Centers for Disease Control and Prevention (CDC), approximately 48 million cases of foodborne illness occur annually in the United States (US).”

- L60-62: This information should not be accurate…non-typhoidal *Salmonella enterica* is the second leading cause of foodborne illness, and not specifically the serotype Typhimurium…

- L74: correct to “…critical control points”

- L163: revise to “…in produce (i.e. shredded lettuce) wash water acquired from…”

- L201: correct to “It was anticipated that droplet integrity could be…”

- L203: the title of Table 1 should be revised for syntax; Suggestion: “Generation time and observed lag time of the tested bacteria in different culture broth media (TSB, BPW and RV broth) at 37°C or 41.5°C (RV broth only), with an initial bacterial concentration of 10^3^  CFU/ml.”

- Also, what do you mean by “observed lag”? Why primary modelling wasn’t attempted so that an actual estimate of lag phase duration to be attained?

- L208: please correct “medias” to media”; singular: medium vs. plural: media; similarly throughout the text of the manuscript. For instance in L354: correct to “a fluorescent medium”

- L300: change to “spread on TSA plates”

- L302: please correct to “effects…were not examined”

- L328: a growth rate cannot be restricted…a growth rate can be reduced

- L337: a broth medium (RV) cannot have a superior growth rate since a broth medium cannot grow…please be careful with the wording used throughout the manuscript. In this particular case, the appropriate wording would be something like “As demonstrated by the collected data, a more prolific bacterial growth was supported during incubation in RV broth at 37°C as compared to 41.5°C”

- L399-400: *S.* Typhimurium is a serotype, not a strain…

- 580: Do not capitalize in table titles

- L585: please provide all pertinent information for the automated turbidimetric system Bioscreen C, namely model, manufacturer, city, country. Act similarly for all materials and equipment described in the manuscript.

- L587: correct “growing temperature” to “growth temperature”; similarly wherever else applicable throughout the text of the manuscript.

We would appreciate receiving your revised manuscript by Apr 04 2020 11:59PM. To enhance the reproducibility of your results, we recommend that if applicable you deposit your laboratory protocols in protocols.io, where a protocol can be assigned its own identifier (DOI) such that it can be cited independently in the future. For instructions see: http://journals.plos.org/plosone/s/submission-guidelines#loc-laboratory-protocols

We look forward to receiving your revised manuscript.

Kind regards,

Alexandra Lianou, Ph.D.

Academic Editor

PLOS ONE

Journal Requirements:

3. Please ensure that you refer to Figure 9 in your text as, if accepted, production will need this reference to link the reader to the figure.

Reviewers' comments:

Reviewer's Responses to Questions

**Comments to the Author**

1. Is the manuscript technically sound, and do the data support the conclusions?

Reviewer #1: Yes

2. Has the statistical analysis been performed appropriately and rigorously? 

Reviewer #1: Yes

3. Have the authors made all data underlying the findings in their manuscript fully available?

Reviewer #1: Yes

4. Is the manuscript presented in an intelligible fashion and written in standard English?

Reviewer #1: Yes

5. Review Comments to the Author

Reviewer #1: The manuscript titled “Microfluidic droplet application for bacterial surveillance in fresh-cut produce wash waters” reports a series of experiments aimed at optimizing the use of droplets for selective screening of Salmonella typhimurium bacteria from wash waters used in industrial food processing.

While microfluidic bacterial detection platforms are not particularly novel, this paper is nevertheless a worthwhile demonstration of the technology being used for analyzing field samples, which in this case comes from fresh-cut produce wash water. In terms of the background and motivation, the manuscript is convincingly written to convey the relevance of the reported work for public health and the utility of emergent droplet microfluidic technologies for this purpose. Moving forward to the results and discussion section, the logical flow of the experiments performed is sound, culminating in a culture-based setup capable of rapid and accurate detection for S. typhimurium. The technical advances from the reported work include rapid detection time (i.e. < 5 hours) and high selectivity in both isolating and visualizing S. typhimurium versus non-S. typhimurium species from field sampled water.

Nevertheless, there are some lapses of clarity that require attention. In the paragraph beginning in line 208, it is unclear if the ‘intensity of fluorescence’ in line 210 refers to media auto-fluorescence or the fluorescence intensity arising from FITC-labelled antibodies in solution. Moreover, the lack of an illustration or schematic in the results section leaves readers guessing how the droplets that encapsulate bacterial cells are produced in the first place. It would be more helpful to move the workflow diagram in Figure 9 from line 685 in the Materials and Methods section to line 208 in the Results section since all subsequent results are derived from the production of these droplets.

Once these concerns have been addressed, I give my approval for this manuscript to be published on this journal as it is an illustrative case study in the area of droplet microfluidic bacterial detection technology.

6. PLOS authors have the option to publish the peer review history of their article (what does this mean?). If published, this will include your full peer review and any attached files.

Reviewer #1: No

---

## [Author Response · Author response to Decision Letter 0]

22 Apr 2020

Ms. Ref. No.: PONE-D-19-34504

Dear Dr. Lianou,

Thank you for the opportunity to respond to the insightful comments you provided as they have resulted in what we feel is an even stronger manuscript. We have responded to each comment below in blue text. We have also indicated where in our revised manuscript we have made modifications in the “Revised Manuscript with Track Changes.” 

Academic Editor comment: The “Introduction” section is very long for a research paper; only in review articles is justified for such section to be wordy and actually include subsections. Please reduce significantly the length of the introduction and revise its content so that excessive details are avoided and a focus is placed on the information providing the rationale for conducting this particular study. Some of the information currently present in the introduction would better fit in the “Discussion” section of the manuscript, with the latter, however, also being rather wordy and requiring significant revision.

We thank the Academic Editor for their comments. In response, the “Traditional culture versus culture independent methods” subsection in the Introduction and “Potential impacts for smaller farms and national policy” subsection in the Discussion have been deleted. In addition to refining the Introduction and Discussion sections, the Introduction subsection headings have been removed. These deletions can be identified in the “Revised Manuscript with Track Changes” and are reflected in the revised “Manuscript.”

L24: please revise to “ready-to-eat produce commodities”

We thank the Academic Editor for this revision. The revision can be identified in the “Revised Manuscript with Track Changes” and is reflected in the revised “Manuscript.”

L27: revise to “Salmonella enterica ser. Typhimurium” (since it is the first time the bacterial species is mentioned and its full name should be presented); upon first mentioning, you can directly use “S. Typhimurium” for the rest of the manuscript. Please make sure that genus and species are in italics, whereas serotype is capitalized and not in italics.

We thank the Academic Editor for these revisions. The revisions can be identified in the “Revised Manuscript with Track Changes” and is reflected in the revised “Manuscript.”

L31 and throughout the text of the manuscript: use “h” instead of “hours”

We thank the Academic Editor for this revision. The revision can be identified in the “Revised Manuscript with Track Changes” and is reflected in the revised “Manuscript.”

L34: please revise to “4-h incubation”

We thank the Academic Editor for this revision. The revision can be identified in the “Revised Manuscript with Track Changes” and is reflected in the revised “Manuscript.”

L40: correct to “…incubated for 4 h…”

We thank the Academic Editor for this revision. The revision can be identified in the “Revised Manuscript with Track Changes” and is reflected in the revised “Manuscript.”

L35-38: are these numbers referring to the relative fluorescence intensity? Please rephrase for clarification.

We thank the Academic Editor for their comment. These below revision can be identified in the “Revised Manuscript with Track Changes” and is reflected in the revised “Manuscript.”

Original text: “… S. typhimurium = 2.36 (95% CI: 2.15-2.58), Enterobacter aerogens 1.12 (95% CI: 1.09-1.16), Escherichia coli 700609 = 1.13 (95% CI: 1.09-1.17), E. coli 13706 1.13 (95% CI: 1.07-1.19), E. coli 700891 1.05 (95% CI: 1.03-1.07) and Citrobacter freundii 1.04 (95% CI: 1.03-1.05).”

Revised text: “… relative fluorescence intensity for S. Typhimurium = 2.36 (95% CI: 2.15-2.58), Enterobacter aerogens 1.12 (95% CI: 1.09-1.16), Escherichia coli 700609 = 1.13 (95% CI: 1.09-1.17), E. coli 13706 1.13 (95% CI: 1.07-1.19), E. coli 700891 1.05 (95% CI: 1.03-1.07) and Citrobacter freundii 1.04 (95% CI: 1.03-1.05).”

L53-55: please consider revising to “According to the Centers for Disease Control and Prevention (CDC), approximately 48 million cases of foodborne illness occur annually in the United States (US).”

We thank the Academic Editor for their revision comment. These below revision can be identified in the “Revised Manuscript with Track Changes” and is reflected in the revised “Manuscript.”

Original text: The United States (U.S.) Centers for Disease Control and Prevention (CDC) estimates there are approximately 48 million cases of foodborne illness in the U.S. each year.

Revised text: According to the Centers for Disease Control and Prevention (CDC), approximately 48 million cases of foodborne illness occur annually in the United States (US).

L60-62: This information should not be accurate…non-typhoidal Salmonella enterica is the second leading cause of foodborne illness, and not specifically the serotype Typhimurium…

We thank the Academic Editor for their revision. These below revision can be identified in the “Revised Manuscript with Track Changes” and is reflected in the revised “Manuscript.”

Original text: “… non-typhoidal S. typhimurium is the second leading cause of foodborne illness in the U.S. with over one million cases per year and is estimated to cause more hospitalizations and deaths than any other foodborne pathogen.”

Revised text: “… non-typhoidal Salmonella enterica is the second leading cause of foodborne illness in the U.S. with over one million cases per year and is estimated to cause more hospitalizations and deaths than any other foodborne pathogen.”

L74: correct to “…critical control points”

We thank the Academic Editor for this revision. The revision can be identified in the “Revised Manuscript with Track Changes” and is reflected in the revised “Manuscript.”

L163: revise to “…in produce (i.e. shredded lettuce) wash water acquired from…”

We thank the Academic Editor for this revision. The revision can be identified in the “Revised Manuscript with Track Changes” and is reflected in the revised “Manuscript.”

Original text: “… identifying S. typhimurium in a produce wash water acquired from a real-world Producer.”

Revised text: “… identifying S. Typhimurium in produce (i.e. shredded lettuce) wash water acquired from a real-world Producer.”

L201: correct to “It was anticipated that droplet integrity could be…”

We thank the Academic Editor for this revision. The revision can be identified in the “Revised Manuscript with Track Changes” and is reflected in the revised “Manuscript.”

Original text: It was anticipated droplet integrity could be compromised at prolonged exposure to 41.5�C.

Revised text: It was anticipated that droplet integrity could be compromised at prolonged exposure to 41.5�C.

L203: the title of Table 1 should be revised for syntax; Suggestion: “Generation time and observed lag time of the tested bacteria in different culture broth media (TSB, BPW and RV broth) at 37°C or 41.5°C (RV broth only), with an initial bacterial concentration of 103 CFU/ml.” Also, what do you mean by “observed lag”? Why primary modelling wasn’t attempted so that an actual estimate of lag phase duration to be attained?

We thank the Academic Editor for this revision. The revision can be identified in the “Revised Manuscript with Track Changes” and is reflected in the revised “Manuscript.” Moreover, primary modelling was not carried out because it was not considered essential to answer the research question at the time.

Original text: Doubling timea and observed lag timeb with a starting cell concentration of 103 CFU/ml in TSB, BPW, and RV broth media grown at 37�C or 41.5�C (RV only)

Revised text: Generation timea and lag phase durationb of the tested bacteria in different culture broth medium (TSB, BPW and RV broth) at 37°C or 41.5°C (RV broth only), with an initial bacterial concentration of 103 CFU/ml medium

L208: please correct “medias” to media”; singular: medium vs. plural: media; similarly throughout the text of the manuscript. For instance in L354: correct to “a fluorescent medium”

We thank the Academic Editor for these revisions. The revisions can be identified in the “Revised Manuscript with Track Changes” and is reflected in the revised “Manuscript.”

L300: change to “spread on TSA plates”

We thank the Academic Editor for this revision. The revision can be identified in the “Revised Manuscript with Track Changes” and is reflected in the revised “Manuscript.”

Original text: “… PFO releasing the trapped incubated content, diluted accordingly, spread to TSA plates, and incubated over night at 37�C.”

Revised text: “… PFO releasing the trapped incubated content, diluted accordingly, spread on TSA plates, and incubated over night at 37�C.”

L302: please correct to “effects…were not examined”

We thank the Academic Editor for this revision. The revision can be identified in the “Revised Manuscript with Track Changes” and is reflected in the revised “Manuscript.”

Original text: Given PFO was introduced upon completion of incubation in the droplet, effects of PFO on bacterial growth was not examined.

Revised text: Given PFO was introduced upon completion of incubation in the droplet, effects of PFO on bacterial growth were not examined. 

L328: a growth rate cannot be restricted…a growth rate can be reduced

We thank the Academic Editor for this revision. The revision can be identified in the “Revised Manuscript with Track Changes” and is reflected in the revised “Manuscript.”

Original text: However, the growth rate of S. typhimurium was substantially restricted at 41.5�C, when compared to S. typhimurium grown in RV at 37�C (Table 2).

Revised text: However, the growth rate of S. Typhimurium was substantially reduced at 41.5�C, when compared to S. Typhimurium grown in RV at 37�C (Table 2).

L337: a broth medium (RV) cannot have a superior growth rate since a broth medium cannot grow…please be careful with the wording used throughout the manuscript. In this particular case, the appropriate wording would be something like “As demonstrated by the collected data, a more prolific bacterial growth was supported during incubation in RV broth at 37°C as compared to 41.5°C”

We thank the Academic Editor for this revision. The revision can be identified in the “Revised Manuscript with Track Changes” and is reflected in the revised “Manuscript.”

Original text: These data indicate RV at 37�C has a superior growth rate, when compared to RV at 41.5�C.

Revised text: As demonstrated by the collected data, a more prolific bacterial growth was supported during incubation in RV broth at 37°C as compared to 41.5°C.

L399-400: S. Typhimurium is a serotype, not a strain…

We thank the Academic Editor for this revision. The revision can be identified in the “Revised Manuscript with Track Changes” and is reflected in the revised “Manuscript.”

Original text: It has already been noted S. typhimurium would likely out compete another strain of bacteria in a mixed culture – at least the bacterial strains researched in the study.

Revised text: It has already been noted S. Typhimurium would likely out compete another serotype of bacteria in a mixed culture – at least the bacterial serotypes researched in the study.

L580: Do not capitalize in table titles

We thank the Academic Editor for these revisions. The revisions can be identified in the “Revised Manuscript with Track Changes” and is reflected in the revised “Manuscript.”

L585: please provide all pertinent information for the automated turbidimetric system Bioscreen C, namely model, manufacturer, city, country. Act similarly for all materials and equipment described in the manuscript.

We thank the Academic Editor for these revisions. The following has been added to the manuscript: BioScreen C, Automated Microbiology Growth Curve Analysis System, Growth Curves USA, New Jersey, United States. The revisions can be identified in the “Revised Manuscript with Track Changes” and is reflected in the revised “Manuscript.”

L587: correct “growing temperature” to “growth temperature”; similarly wherever else applicable throughout the text of the manuscript.

We thank the Academic Editor for these revisions. The revisions can be identified in the “Revised Manuscript with Track Changes” and is reflected in the revised “Manuscript.”

Response to Reviewers' Comments:

Dear Reviewer #1,

Thank you for responding with your objective feedback and insightful comments as they have resulted in what we feel is an even stronger manuscript. We have responded to your comment below in blue text. We have also indicated where in our revised manuscript we have made modifications in the “Revised Manuscript with Track Changes.”

Reviewer #1

The manuscript titled “Microfluidic droplet application for bacterial surveillance in fresh-cut produce wash waters” reports a series of experiments aimed at optimizing the use of droplets for selective screening of Salmonella typhimurium bacteria from wash waters used in industrial food processing.

While microfluidic bacterial detection platforms are not particularly novel, this paper is nevertheless a worthwhile demonstration of the technology being used for analyzing field samples, which in this case comes from fresh-cut produce wash water. In terms of the background and motivation, the manuscript is convincingly written to convey the relevance of the reported work for public health and the utility of emergent droplet microfluidic technologies for this purpose. Moving forward to the results and discussion section, the logical flow of the experiments performed is sound, culminating in a culture-based setup capable of rapid and accurate detection for S. typhimurium. The technical advances from the reported work include rapid detection time (i.e. < 5 hours) and high selectivity in both isolating and visualizing S. typhimurium versus non-S. typhimurium species from field sampled water.

Nevertheless, there are some lapses of clarity that require attention. In the paragraph beginning in line 208, it is unclear if the ‘intensity of fluorescence’ in line 210 refers to media auto-fluorescence or the fluorescence intensity arising from FITC-labelled antibodies in solution. Moreover, the lack of an illustration or schematic in the results section leaves readers guessing how the droplets that encapsulate bacterial cells are produced in the first place. It would be more helpful to move the workflow diagram in Figure 9 from line 685 in the Materials and Methods section to line 208 in the Results section since all subsequent results are derived from the production of these droplets.

Once these concerns have been addressed, I give my approval for this manuscript to be published on this journal as it is an illustrative case study in the area of droplet microfluidic bacterial detection technology.

We thank Reviewer #1 for these comments and recommended revisions. The revisions can be identified in the “Revised Manuscript with Track Changes” and is reflected in the revised “Manuscript.” We moved Figure 9 up in the manuscript to Figure 1 – aligning well with the “Proof-of-concept experimental design” subsection and the beginning of the Results section. We have tried to correct the lapses in clarity for the identified section.

---

## [Editor Report · Decision Letter 1]

1 May 2020

Microfluidic droplet application for bacterial surveillance in fresh-cut produce wash waters

PONE-D-19-34504R1

Dear Dr. Harmon,

We are pleased to inform you that your manuscript has been judged scientifically suitable for publication and will be formally accepted for publication once it complies with all outstanding technical requirements.

With kind regards,

Alexandra Lianou, Ph.D.

Academic Editor

PLOS ONE

Additional Editor Comments (optional): The comments raised during the review of the original submission have been adequately and sufficiently addressed by the authors. No further comments are raised.
---

## [Editor Report · Acceptance letter]

18 May 2020

PONE-D-19-34504R1 

Microfluidic droplet application for bacterial surveillance in fresh-cut produce wash waters 

Dear Dr. Harmon:

I am pleased to inform you that your manuscript has been deemed suitable for publication in PLOS ONE. Congratulations! Your manuscript is now with our production department. 

With kind regards,

on behalf of

Dr. Alexandra Lianou 

Academic Editor

PLOS ONE